# Risk Early Warning of Food Quality Safety in Meat Processing Industry

**DOI:** 10.3390/ijerph17186579

**Published:** 2020-09-09

**Authors:** Yifan Xiong, Wei Li, Tiezhong Liu

**Affiliations:** School of Management and Economics, Beijing Institute of Technology, Beijing 100081, China; 3120191571@bit.edu.cn (Y.X.); liweibeijingligong@163.com (W.L.)

**Keywords:** early warning, food safety, risk management, risk indexes

## Abstract

In recent years, people’s demand for meat products has increased, but the occurrence of meat food quality and safety problems has also caused irreparable losses to the safety of human lives and properties, and enterprises have lost their reputation. Since the frequent occurrence of food quality and safety incidents is the result of the lack of an early warning mechanism, a large number of problematic foods flow into the market. In order to prevent the occurrence of food quality and safety incidents and control food quality from the source, this article first refers to the results of EFSA’s Emerging Risks Project (EMRISK) and the food safety early warning framework of Kleter and Marvin, combined with the existing meat processing companies. Some quality control systems have put forward an early warning indicator system that includes the external environment of the enterprise, internal risks, and consumers’ concerns. Then, by issuing 500 questionnaires and interviewing 25 experts, 912 pieces of data were collected and a Monte Carlo simulation early warning model was established. Using case studies, taking Shandong Delis Co., Ltd. (Binzhou, China, hereinafter referred to as DLS) as an example, through sensitivity analysis and program analysis, the company’s food risk status and early warning model was evaluated. The results show that the risk of rising consumers’ concerns about counterfeiting and inferior products has the greatest impact on food quality and safety risks, followed by policy adjustment risks, and the risk of raw material sources ranked third. A total of six important risk warning indicators have been extracted, and these six need to be strictly controlled to control the overall risk. The research provides support for companies to formulate food quality monitoring, early warning and management strategies from a macro perspective, and control key early warning indicators in food quality and safety to reduce risks.

## 1. Introduction

Food is the basic element to maintain human survival, activities, and development. With the progress of society, food safety has attracted more and more attention from consumers [1]. However, the current food safety situation in our country is not optimistic. Factors such as highly toxic pesticides, harmful chemical substances and microbial contamination, and the dishonest behavior of food producers directly affect the safety of food. Food safety issues have caused endless disasters [2]. According to the survey provided by the World Health Organization, about 600 million cases of food-borne diseases occur every year, and unsafe food poses a threat to human health and the economy. Therefore, in order to effectively prevent the occurrence of food quality and safety problems, it is very important to establish a food quality risk analysis framework and determine key risk control factors.

In terms of food quality and safety risk identification, scholars at home and abroad mainly study three aspects: The steps to identify food safety risks, the systems and methods for identifying food safety risks, and the types of food safety risks. Ariane has established a food safety risk analysis framework structure, including framework concept construction, risk-benefit assessment, evaluation and risk management. Wentholt and Fischer used two international rounds of Delphi to effectively identify food risks. Experts involved in identifying risks include scholars from different countries who are responsible for formulating different laws and regulations. They also recommend international data sharing and adopt forward-looking methods to urge developing countries, improve risk identification, and control capabilities [3]. Marvin divided the strategies for identifying risks in advance into three types: Reaction systems, predictive and early warning systems, and systems based on historical data [4]. Kleter and Marvin believed that food quality and safety hazards should be found from multiple angles, including the environment of food production, the process of food from the place of production to the table, and the three aspects of consumers [5], etc. [6]. Petersen and Knura-Deszczka conducted process and node data collection and processing on systematic and regular behaviors in each stage of production, mainly using checklists, sorting, monitoring suppliers, and inspecting the sanitary environment to discover hazards [7]. Hou believed that food quality and safety should be analyzed from the aspects of agricultural product raw materials, processing and manufacturing, market and circulation [8]. Wang believed that food safety problems are caused by source pollution, heavy metal pollution, biotoxins, genetically modified foods, food adulteration, and poisoning [9]. Pang and Li pointed out that food safety problems mainly arise from three aspects: Infection or poisoning, food-borne diseases caused by factors that enter the body through the digestion of food; biological and chemical pollutant hazards; new food technologies and new resources (such as genetically modified uncertainty caused by food, enzyme preparations, and new food packaging materials) [10].

Some scholars have conducted research on risk control systems and early warning systems, among which HACCP has attracted the attention of many scholars: Hartog creatively applied HACCP to animal feeding, connecting the feeding link with the food production and processing link [11]. Asselt and Meuwissen paid attention to the selection of key control points in HACCP. They looked back on the external factors, producer factors, technological innovations, internal factors, consumer factors, vision, etc. in the past six years, and compared them. They found significant changes in factors, and then compared them. These major changes are linked to food safety hazards, so as to discover risks and determine critical control points [12]. Janevska and Gospavic used HACCP and Quantitative Microbiological Risk Assessment (QMRA) to study the impact of climate change on food quality and safety [13]. This article will use HACCP to study food quality and safety.

Food quality and safety risk early warning can be divided into three types: Single indicator early warning, comprehensive early warning, and a combination of the two. The domestic research results are mainly reflected in two categories. The first category is the research results of Tang [14] and Li [15]. Tang [14] established a multi-alert multi-level parallel structure, which is divided into three levels: Overall level, system level, and index level. The system level is divided into food quantity safety, quality safety, and sustainable safety. The second category is the indicator system of Liu [16] and Huang [2], including three levels of project indicators, food category indicators, and overall status indicators. In this paper, referring to the opinions of domestic and foreign scholars, the factors affecting the food quality and safety of manufacturing and processing enterprises can be divided into three parts: Enterprise external environmental risks, enterprise internal risks, and consumer risks.

The innovations of this article are: First of all, the theoretical research and practical discussion of China’s food quality and safety risk early warning system have so far mainly focused on the national, government, and regional regulatory levels. There are not many food quality and safety risk early warning methods for the enterprises’ own use, especially the application of quality and safety early warning for meat processing enterprises is not yet mature, and most of them are limited to a single indicator early warning. Therefore, the establishment of early warning indicators and early warning systems for meat processing enterprises can help them in their food quality and safety management. Secondly, in addition to export companies and large-scale food production companies, domestic sales companies and small and medium-sized companies have not yet fully applied HACCP. One of the main reasons is insufficient risk analysis capabilities and inaccurate positioning of key control points. It is very valuable to establish a risk analysis system and method to integrate it with HACCP to make HACCP truly feasible. Finally, the Monte Carlo method is less applied in the research of food quality and safety risk warning, and is limited to a single-factor simulation of exposure assessment of hazardous substances. This is a brand new attempt to be used in the quality and safety risk early warning research of large-scale meat processing enterprises.

## 2. Methodology

### 2.1. The Risk Early Warning Index 

The risk early warning index is the basis of risk early warning system. According to GB/T 4754-2002, the meat processing industry includes mature meat products processed from livestock and poultry, and each link of the meat producing chain may cause quality safety risks. 

Food quality safety risks are caused by both internal and external factors of the food chain. The internal factors include food chain participants, production links, enterprise scale, technological innovation, etc. External factors include sources of raw materials, laws and regulations, climate impacts, and economic conditions. As a special external factor, the consumer factor is specially emphasized, including the attention to the environment, the attention to animal welfare, the attention to sensory quality, convenience, and health [12]. Kleter and Marvin also believed that hazards should be looked for from multiple perspectives, including the external environment, the process of food production from place to table, and consumers [17]. Among them, hazard factors in the external environment are mainly identified by EMRISK. HACCP and RASFF were used to identify the hazards from the production site to the dinner table. In terms of consumers, the consumers’ complaints and monitoring of food safety incidents are identified.

Then, it can be seen from the literature (EFSA, Kleter and Marvin) that the factors affecting the food quality safety can be divided into three parts: External environmental risk, internal environmental risk, and risk of consumers concern. Thus, the risk early warning index was obtained as in Table 1.

#### 2.1.1. External Environmental Risk

External environment will directly or indirectly affect the producing process, and then affect food quality safety. Based on the previous literature, the external environment risk is divided into four dimensions.

(1) Policy adjustment risk 

The policy adjustment risk includes two aspects: The first is the absence of regulation and policy, i.e., even though the progress of human society, the development of Science and Technology, the variety and nature of food have led the original food standard policy as inapplicable, there might be still no new standard put forth. The second aspect refers to the increase of international trade barriers, or government making adverse policies to the trade enterprises.

The shortage of standard will make enterprises unable to find enough reference for the manufacturing process, resulting in uneven or potentially unsafe products. For example, because of China’s shortage of standards on food additives, many additives have since not been detected or are subjected to no detection requirements, which can lead to the occurrence of Clenbuterol, melamine, and other incidents.

(2) Economic environment risk

The bad economic environment means that the social and economic conditions for enterprises’ survival and development become worse, for either the national economic policy is unfavorable to the production and development of enterprises, or the purchasing power of consumer decreases. A bad economic environment will reduce the enterprise’s investment on food quality safety, which can affect the food quality safety.

(3) Natural disasters and climate impacts

The influence of natural disasters and climate change on food quality safety is obvious and immediate, that is, the producing will either be stopped or delayed, making food either polluted or mildewed. For example, changes in temperature and humidity can affect the growth of fungi, which in turn can affect the production of mycotoxins [17]. As a result of climate warming, the toxicity of microorganisms and the mode of transmission will change also, making the originally harmless substances gradually become harmful to the human body, or bringing microorganisms that once disappeared from the food production and circulation back to the chain. These effects and changes may be imperceptible over a period of years or even decades, but that does not mean they will be less harmful.

(4) Source hazards of raw materials

Source hazards of raw materials refer to the food safety hazards that may be brought into the manufacturing process by raw materials purchased from the frontline of the supply chain, including biological, chemical, and physical pollution, and allergens. EFSA’s emerging risk factors the “retail trade”, in particular, mentioned the impact of enterprise procurement on food quality safety, which may bring into enterprises internal hazards of uncertainty [17]. Even though the enterprises can promise on their own production process and product standards and safety, it cannot fully guarantee the safety of raw materials, as their in-plant testing on raw materials before they enter the producing chain is very likely to fail, resulting in the safety risk coming out of ex-factory products.

#### 2.1.2. Internal Environmental Risk

Internal risk factors are mainly extracted by comparing the four systems and certifications that meat manufacturing and processing enterprises need to pass, including ISO 22000:2005, HACCP system, BRC certification, and IFS certification.

(1) Insufficiency of top management responsibility 

Management shall demonstrate support for the food safety objectives of the organization, formulate food safety guidelines, communicate food related laws and regulations to the organization, focus on customers, and ensure regular review of the quality management system (ISO22000, 2005).

Inadequate or inadequate food safety policy. While the food safety policy can be seen as the wind vane of the enterprise, how the shortage of it or improper application can cause the management goal of the enterprise on food quality safety is still not clear.

Unclear organizational structure and job responsibilities. The unclear organization structure and post responsibility in the enterprise will lead to unclear power, responsibility, enterprise’s policy, and quality objectives that will not be able to be carried out by all staff. In particular, if the staff is engaged in a post that has no clear job responsibilities, it will inevitably affect the quality of work, and may lead to food safety risks.

Poor communication. Poor external communication will affect the enterprises to obtain information on food safety, including changes in regulations, raw materials, market changes, customer satisfaction, etc. It will affect the enterprise’s manufacturing process, the direction of standard adjustment and timeliness. Poor internal communication will lead to poor food safety management from top to bottom. While management and operation blind spots can lead to an outbreak on food quality safety risks, a poor communication with the food safety team will affect the updating efficiency of the food safety management system.

(2) Personnel management risk 

The personnel safety awareness and training are only mentioned in ISO22000 and IFS, while job control has been mentioned in the BRC in view of its importance.

Shortage of safety awareness among personnel. Safety awareness plays a decisive role in the behavior of employees, while it inhibits unsafe behavior and special risk-taking behavior. The safety consciousness is not strong, to the danger cognition ability shortages, extremely easy to take risks foolishly, causes the quality safety risk.

Inadequate training of personnel. Shortage of personnel training may lead to manufacturing workers that do not have the required work capacity, directly affecting the quality of products. Especially if responsible for the food safety management system the personnel training is insufficient, that will make the food quality safety system become a dead letter.

Wrong operation of the personnel. Although many production processes are machine-operated, there are still many processes that need manual operation, such as weighing the raw materials before the mixing process, disoperation of the staff may lead to inappropriate mix, and product nutrition index is not qualified.

(3) Hazard control risk 

The aim of hazard control is to control, and the limit of low risk is the minimum detectable quantity. The alert limits of medium, low, medium and high risks are classified according to the literature, expert opinions, and enterprise standard documents.

(1) Chemical Hazard

Animal residue—Doxycycline. Doxycycline is one of the main index of animal residues detected by meat processing enterprises. The classification of warning levels and warning limits were determined according to the maximum residue limits (mrls) for foods of animal origin 2377/90/EC.

Animal residue—Enrofloxacin. Enrofloxacin is currently designated by the state as a special drug for animals, and it has a long half-life inside the animal body. Its good tissue distribution makes it a broad-spectrum bacteriostatic agent for Gram-positive bacteria, negative bacteria, and mold paste. Its bacteriostatic effect has been used in the control of vibriosis and colibacillosis in cultured fish. The classification of warning levels and warning limits were determined according to the maximum residue limits for foods of animal origin 2377/90/EC.

Drug residue—Chlorpyrifos. Soluble in most organic solvents and of moderate toxicity. Determination on its biological, chemical, and physical hazards in meat processing were made according to maximum residue limits of pesticides in foods GB 2763-2005. 

Drug residue—Flumetasone. Insecticide, moderate toxicity, that are mainly used in dogs, cat surface fleas, dog ticks, and other surface pests. Determination of early warning classification and warning limits were made based on maximum residue limits of pesticides in foods GB 2763-2005.

Nitrite. The probability of food poisoning caused by nitrite is high. The ingestion of 0.3~0.5 g nitrite can cause poisoning or even death. Nitrite is also a carcinogen. There is a positive correlation between esophageal cancer and intake of nitrite. The classification of warning grade and warning limit were determined based on the hygienic standard of nitrite limit and the prevention of nitrite food poisoning. 

Heavy metal—Mercury. Trace amounts of mercury do not cause harm in the human body and can be excreted through urine, feces, and sweat. However, if the amount is too much, it can damage human health. The early warning level classification and warning limit were determined according to the comparison and analysis of China and EU food heavy metal limit standards. 

Heavy metal—Cadmium. When the environment is polluted by cadmium, cadmium can be enriched in organisms and enter the human body through the food chain, causing chronic poisoning. Even though the cadmium can be absorbed by the human body, the cadmium thiophane protein will be formed in the body, and will be selectively accumulated within livers or kidneys. The kidney, as it can absorb nearly 1/3 of cadmium inside body parts, has long been held as the target organ for cadmium poisoning. Other organs such as the spleen, pancreas, thyroid, and hair also have a certain amount of accumulation. The early warning level classification and warning limit are determined according to the comparison and analysis of China and EU food heavy metal limit standards. 

Heavy metal—Lead. Lead is a toxic metal, long-term exposure to lead and its salts can cause kidney disease and colic-like abdominal pain. Excessive intake of lead and its compounds can lead to palpitations, irritability, and damage to the nervous system. It can even cause cancer and deformities. Excessive levels of lead can have a very negative effect on children. It can damage the nervous system of children and lead to diseases of the blood circulation and brain. Lead can accumulate in the human body and is very difficult to eliminate automatically. It can only be removed through certain drugs [18]. The early warning level classification and warning limit were determined according to the comparison and analysis of China and EU food heavy metal limit standards. 

(2) Biological hazard

Colony count. The colony count includes pathogenic bacteria and beneficial bacteria. The harmful bacteria to the human body mainly belong to pathogenic bacteria. If the total number of bacterial colony exceeds the standard, it also can increase the chance of pathogenic bacteria exceeding the standard, increasing the risk of harm to human health. If the total number of bacteria in food exceeds the standard seriously, it will indicate that the hygienic condition of food cannot meet the basic hygienic requirement, and will eventually destroy the nutritional composition of food, accelerating the spoilage of food and making the food inedible. Consumers who eat those food in which the microorganism exceeds the standard quite a lot, are very easy to suffer from dysentery and other intestinal diseases, such as vomiting, diarrhea, etc., thus endangering the human body health safety. Classification of warning levels and warning limits were determined according to the literature on contamination and risk assessment of pathogenic microorganisms.

Coliform bacteria. It is generally believed that this group of bacteria can include Escherichia coli, Citrobacter, Gas-producing Klebsiella, and Enterobacter cloacae, etc. If there is fecal contamination in food, it is speculated that there should be a possibility of contamination of intestinal pathogenic bacteria in food, and the threat of food poisoning and epidemics is lurking, which must be regarded as potential harm to human health. The classification of warning level and warning limit were determined according to the literature of contamination and risk assessment of pathogenic microorganisms. 

(3) Physical hazard

Iron (Fe). Metal detection is used as a key detector in most HACCP systems for meat processing companies. The objective is to reduce or eliminate the risk of metal foreign objects contaminated inside the product and get the objects effectively removed. Early warning classification and warning limits were determined according to the US FDA, Japanese, and EU standards.

Stainless steel (SUS). SUS is the code name for stainless steel in the Japanese JIS standard. Early warning classification and warning limits were determined according to the US FDA, Japanese, and EU standards.

(4) Product innovation risk

Changes in new products or related products, packaging materials, or manufacturing processes in the enterprise shall have appropriate product design/development procedures to ensure the change is legal and safe on all new products, including formulations and packaging for the existing products. Any change in the process should be approved by the HACCP team leader or authorized HACCP committee members (ISO22000, 2005).

The product innovation, including product, packaging, transportation methods, and process innovations poses a risk to food safety. On one hand, the innovation of products and processes means unknown hazards and risks, on the other hand, the new products and processes can exceed existing standards. Thus, the shortage of reference is also an important cause for food quality safety risks.

#### 2.1.3. Risk of Consumers Concern

Consumers’ attention can directly affect the organization’s attention. For example, if consumers pay more attention to sensory quality, companies will need to make further improvements and efforts in the sensory quality of products.

(1) Concerned about the harmful substances

The harmful substances of the product exceed the standard, that is, the detection indexes such as the total number of colonies, aflatoxin, Escherichia coli, sweetener, etc. exceed the national limit, or excessively add harmful additives to the human body. Due to the continuous improvement of consumers’ awareness on consumer rights, consumers will pay extensive attention to the content of harmful substances in products. Once the harmful substances of products are exposed, the operations of the enterprises and manufacturers involved will be greatly affected, and may even force the companies to file bankruptcy or seek acquisition.

(2) Concerned about counterfeiting

Counterfeit and shoddy products are counterfeit, inferior, false, and counterfeit. Mainly reflected in the quality products to pretend to be high-quality products, exaggerating the function, and nutritional value of products, deceiving consumers.

(3) Concerns about sensory quality

Sensory quality, customers will take the sensory test at the time of purchasing, or known also as “functional test”. It is to rely on the human sensory organs to evaluate and judge the quality of the product. For example, the shape, color, smell, scar, and aging degree of the product are usually checked out by the sensory organs such as human vision, hearing, touch, and smell, with the quality being judged to be either good or bad. Not only consumers, but also professionals often use food quality sensory inspection technology to evaluate food quality in the food production process.

(4) Concerns about the environment

Environment index refers to food production and sales environment, production and sales of personnel hygiene. If the production and sale of the environment is not up to standard, it will cause food pollution and nutritional value decline.

### 2.2. Risk Early Warning Model

#### 2.2.1. Framing

The modeling process with the Monte Carlo simulation method can be divided into three steps:

First, establishing functional. This method needs to input the function of each risk factor as a premise of simulation. If the consequences of risk events Φ and risk variables $\mu_{1,}\mu_{2,\ldots }\mu_{n}$ are related, a functional between Φ and $\mu_{1,}\mu_{2,\ldots }\mu_{n}$ will need to be established [19]:(1)$$\Phi =f\;\left(\mu_{1,}\mu_{2,\ldots }\mu_{n}\right)\;$$

Generally speaking, there is a ready-made NPV or IRR model for Monte Carlo simulation. However, there is no ready-made model, the analytic hierarchy process is adopted with the experts’ judgment. 

Second, determining input variables and probability distribution. That is to analyze and fit the data as well as select the most suitable probability distribution. That is to determine the probability distribution function of the variable $g_{1}\left(\mu_{1}\right),\;g_{2}\left(\mu_{2}\right),\;\ldots g_{3}\;\left(\mu_{n}\right).$

Third, random sampling and simulation. The Latin hypercube sampling method is used to produce a set of accord with the distribution of the random number sequence $\left\{\mu_{1,}\mu_{2,\ldots }\mu_{n}\right\}$. It will get a random sampling sequence $\left\{\mu_{1,}\mu_{2,\ldots }\mu_{n}\right\}$ into $\Phi =f\;\left(\mu_{1,}\mu_{2,\ldots }\mu_{n}\right)$, that can get a Φ risk value. By continuing the use of random number generator for N times sampling, a random N group sequence will be got. By putting the random numbers of N groups into the risk consequence function, the risk value of each of the N random numbers will be calculated $\left\{\Phi_{1},\;\Phi_{2},\;\ldots ,\;\Phi_{n}\right\}.$
(2)$$\Phi_{1}=f\;\left(\mu_{1}^{1},\;\mu_{2}^{1},\;\ldots \mu_{n}^{1}\right)\;\Phi_{2}=f\;\left(\mu_{1}^{2},\;\mu_{2}^{2},\;\ldots \mu_{n}^{2}\right)\;\;\Phi_{3}=f\;\left(\mu_{1}^{3},\;\mu_{2}^{3},\;\ldots \mu_{n}^{3}\right)\;\;\Phi_{n}\;=f\left(\mu_{1}^{n},\;\mu_{2}^{n},\;\ldots \mu_{n}^{n}\right)$$

Each sampling and simulation is random and independent, with enough trials to accurately reflect the distribution characteristics of the function. After statistical analysis, the statistical characteristic quantity (mean value, variance, and its cumulative probability distribution) of the risk consequence is calculated.

#### 2.2.2. Weight Determination

Establish a hierarchical model. The target layer is “food quality safety risk value of meat products manufacturing and processing enterprises”; the middle layer is the first-level index of the risk early warning index: The external environmental risk, internal environmental risk, and risk of consumers concern. There are two layers on layer measures and schemes. One is 12 second-level indicator, and the other is nine third-level indicators.

After the establishment of the hierarchy, it is necessary to determine several elements *x*_1_, *x*_2_, …, *x_m_* that are governed by *z* (except for the bottom layer). Regarding the sorting weights of z, these weights *p*_1_, *p*_2_, …, *p_m_* are often expressed as a percentage, which meet 0 ≤ *p_j_* ≤ 1 and $\overset{m}{\underset{j=1}{\sum }}p_{j}=1$. The judgment matrix is established through the comparison method. Let m elements governed by upper element *z* be *x*_1_, *x*_2_, …, *x_m_*, for i, j=1, 2, …, m, the judgment matrix of $x_{i}$ and $x_{j}$ for the pairwise comparison of *z* is *a_ij_*, which is determined by Satty’s nine scale method.

Suppose the composition weight of $n_{k-1}$ elements in the *k −* 1 layer relative to the target is calculated as:(3)$$\;w^{\;\left(k-1\right)\;}=\;\left(w_{1}^{\left(k-1\right)},\;w_{2}^{\left(k-1\right)},\;\ldots ,\;w_{n_{k}}^{\left(k-1\right)}\right){\;}^{T}$$

Let us also set the weight vector of $n_{k}$ elements in the *k* layer. For the *j**-*th element ($j=1,\;2,\;\ldots ,\;n_{k-1})$ in the *k* − 1 layer, the sorting weight vector is$\;u_{j}^{\;\left(k\right)\;}=\;{\left(u_{1j}^{\left(k\right)},\;u_{2j}^{\left(k\right)},\;\ldots ,\;u_{n_{k}j}^{\left(k\right)}\right)}^{T}\;j=1,\;2,\;\ldots ,\;n_{k-1}$. This weighting vector shall be complete for the *k* layer’s $n_{k}$ elements. When some elements are not dominated by the *j* in the *k* − 1 layer, the corresponding position is supplemented by 0 and then the $n_{k}\times n_{k-1}$ matrix is obtained:(4)$$\;U^{\;\left(k\right)\;}=\left(\begin{array}{llll} U_{11}^{\;\left(k\right)\;} & \;U_{12}^{\;\left(k\right)\;} & \;\ldots \; & U_{1n_{k-1}}^{\;\left(k\right)\;}\\ U_{21}^{\;\left(k\right)\;} & \;U_{22}^{\;\left(k\right)\;} & \;\ldots \; & U_{2n_{k-1}}^{\;\left(k\right)\;}\\ \;\vdots & \;\vdots & & \vdots \\ \;U_{n_{k}1}^{\;\left(k\right)\;} & \;U_{n_{k}2}^{\;\left(k\right)\;} & \;\ldots \; & U_{n_{k}n_{k-1}}^{\;\left(k\right)\;}\; \end{array}\right)$$

Thus, the composition weights of *n_k_* elements in the *k* layer about the target layer can be obtained:(5)$$\;w^{\;\left(k\right)\;}=U^{\;\left(k\right)\;}w^{\;\left(k-1\right)\;}$$

Decomposition can be:(6)$$\;\;w^{\;\left(k\right)\;}=\;U^{\;\left(k\right)\;}U^{\;\left(k-1\right)\;}\ldots U^{\;\left(3\right)\;}w^{\;\left(2\right)\;}\;$$

Therefore, we can get the functional value of food quality safety risk Φ of the target layer:(7)$$\Phi =\overset{n}{\underset{i=1}{\sum }}w_{i}^{\left(k\right)}\mu_{i}=\overset{n}{\underset{i=1}{\sum }}\overset{n_{k-1}}{\underset{j=1}{\sum }}u_{ij}^{\left(k\right)}w_{j}^{\left(k-1\right)}\;(n\;=\;\mathrm{Number}\;\mathrm{of}\;\mathrm{risk}\;\mathrm{factors})\;$$

#### 2.2.3. Probability Distribution

The maximum likelihood estimation method is used to determine the probability distribution of variables. For the discrete population, take the observed sample value $x_{1},\;\ldots ,\;x_{n}$, and write down the probability of occurrence of the observed value, which generally depends on some or some parameters, represented by *θ*, regarding the probability as a function of *θ*, represented by *L*(*θ*), namely:(8)$$L\left(\theta \right)=P\left(X_{1}=x_{1},\;\ldots ,\;X_{n}=x_{n};\theta \right)\;$$

Estimates of the maximum likelihood estimation are to look for *θ*. $\left(\hat{\theta }=\hat{\theta }\;\left(x_{1},\;\ldots ,\;x_{n}\right)\;\right)$ in order to achieve the type of *L*(*θ*).

For a continuous population, let the probability density function of the population be p(x;θ), θ∈Θ, where *θ* is a parameter vector composed of an unknown parameter or several unknown parameters, and is the parameter space where *θ* may take value. $x_{1},\;\ldots ,\;x_{n}\;$ are samples from the population. Consider the joint density function of the sample as a function of theta, using $L\left(\theta ;x_{1},\;\ldots ,\;x_{n}\right)$, shorthanded as *L*(*θ*):(9)$$L\left(\theta \right)=L\left(\theta ;x_{1},\;\ldots ,\;x_{n}\right)=P\;\left(x_{1};\theta \right)\;\cdot P\;\left(x_{2};\theta \right)\;\cdot \ldots \cdot P\;\left(x_{n};\theta \right))$$

*L*(*θ*) is called the likelihood function of the sample. If a particular statistic $\hat{\theta }=\hat{\theta }\left(x_{1},\;\ldots ,\;x_{n}\right)$ contents:(10)$$L\left(\hat{\theta }\right)=max_{\theta \in \Theta }L\;\left(\theta \right)\;$$

$\hat{\theta }\;$ is then called the maximum likelihood estimate of *θ*, abbreviated as MLE.

#### 2.2.4. Tool Selection

Considering that the Latin hypercube sampling can accurately reconstruct the input distribution by sampling with less iteration and making the real statistics of the input distribution faster and more ideally, the Latin hypercube sampling method is adopted. 

The risk analysis software @risk platform is selected for its better processing efficiency and performance on data characteristics. The simulation tool adopts the risk analysis software @risk based on the Monte Carlo simulation technology developed by the American Palisade Company, which is widely used internationally.

@risk introduces the risk analysis technology into the industry-standard spreadsheet software package Microsoft Excel system. With @risk and Excel, risk models can be built. As an add-in to Microsoft Excel, @risk is directly linked to Excel, providing all the necessary tools for setting up, executing, and viewing the results of risk analysis, as well as the processing efficiency is high and it can better express data characteristics, and @risk uses the same style of Excel menus, and functions are more convenient to use.

## 3. Case Study

### 3.1. Date Source

The DLS was selected as an example, which is a large food enterprise that is mainly engaged in pig slaughtering, chilled meat, low-temperature meat products, and prepared food processing. Its products have reached the aseptic level on pure raw edible meat and passed the HACCP certification, the SGS certification of the European Union, and the export certification of Hong Kong, Singapore, and Russia. This firm owns a rich technical capability, including high-quality staff, enterprise laboratories, etc. At the same time, it retains two years of historical data (generally the product shelf life plus one year of storage period), so it can directly go through simulated analysis with the test data. At the same time, the questionnaires can also be used to summarize the opinions of experts. If the number of investigations is large enough, it can even reflect the historical situation. Therefore, combined with research needs, the data sources are divided into two types: Test data and questionnaires.

#### 3.1.1. Routine Testing Data

The hazard items in the hazard control risk part of the internal environmental risk, include chemical hazard, biological hazard, and physical hazard. The corresponding items can be found in the routine testing on the enterprise meat products. Collecting the test data of the enterprise from 2010 to 2012, this can be used as part of the data source for simulation after being processed. The monitoring data of each hazard are very different on both the quantity aspect and on the data unit aspect, it is necessary then to take the dimensionless process on the detection data of each hazard. Considering that the risk degree of various hazards to the human body varies greatly, it has become very difficult to draw clear boundaries for each risk degree of hazards. Therefore, the fuzzy set theory is adopted to blur the risk degree. The risk degree of hazards is divided into five levels: Low risk, medium low risk, medium risk, medium high risk, and high risk. Each class assigns different pollution risk weights *T**_i_* of 10, 30, 50, 70, and 90, respectively.

Although each hazard risk index is divided into five levels, hazard detection data cannot fall exactly on the classification boundary, so a membership degree in the fuzzy mathematical theory is needed to describe the pollution degree. Here, the membership function is determined to be a reduced half trapezoidal distribution [18]. That is, the data record of the *i* hazard indicator detection for the *k* hazard indicator of the *m* pollutant (detection value is *x_i_*), the corresponding membership of the hazard is calculated as:(11)$$h_{i1}=\{\begin{array}{c} 1\;x_{i}\leq q_{k1}\\ \frac{q_{k2}-x_{i}}{q_{k2}-q_{k1}}\\ 0\;x_{i}\geq q_{k2} \end{array}q_{k1}<x_{i}<q_{k2}$$
(12)$$h_{i2}=\{\begin{array}{c} 1-h_{i1}q_{k1}<x_{i}\leq q_{k2}\\ \frac{q_{k2}-x_{i}}{q_{k2}-q_{k3}}q_{k2}<x_{i}<q_{k3}\\ \;0\;x_{i}\geq q_{k3}\;or\;x_{i}\leq q_{k1} \end{array}\;$$
(13)$$h_{ij}=\{\begin{array}{c} 1-h_{i,\;j-1}q_{k,\;j-1}<x_{i}\leq q_{kj}\\ \frac{q_{kj}-x_{i}}{q_{kj}-q_{k,\;j+1}}q_{kj}<x_{i}<q_{k,\;j+1}\\ 0\;x_{i}\geq q_{k,\;j+1}\;or\;x_{i}\leq q_{k,\;j-1} \end{array}\;j=2,\;3,\;\ldots ,\;6$$
where $q_{k,\;j}$ is the cut-off value set by *j* for the risk level of the *k* hazard. Thus, the risk score of the test data record of the *i* article is obtained as follows:(14)$$RV_{ki}=\overset{5}{\underset{j=1}{\sum }}h_{i,\;j}T_{j}\;$$

$h_{i,\;j}\;$ is membership, $T_{j\;}$ is assigning risk weights.

#### 3.1.2. Questionnaire Data

For the external environmental risk, insufficiency of top management responsibility, personnel management risk, product innovation risk, and risk of consumers concerned in the internal environmental risk do not have historical data, the data was obtained through questionnaires. The information of two dimensions of each risk indicator was obtained through the questionnaire: The probability of risk factors occurring in the future p; if the risk breaks out, the degree of impact on the food quality of the enterprise E. The probability of occurrence of the K risk factor is set as *p_k_*, and the influence degree of the occurrence on the food quality of the enterprise is set as *E_k_*. Then, the risk value of the *K* is $RV_{k}=f\left(p_{k},\;E_{k}\right)==p_{k}\times E_{k}$. Considering that the data obtained from the questionnaire are discrete and classified and its contribution to the risk analysis is not very large, the questionnaire is designed only as a continuous line, as an improvement for the Likert scale. Respondents rated the likelihood of risk factors occurring on a scale of 0 to 100% and the degree of impact on a scale of 0 to 10.

The respondents are middle and senior employees of the DLS group, taking three types of positions: Management posts, technical posts, and risk internal audit posts. A total of 500 questionnaires were issued, and 476 valid questionnaires were collected, with an effective questionnaire rate of 95.2%. After the SPSS test, the reliability coefficient Cronbach α was 0.834, indicating that the questionnaire results were acceptable.

### 3.2. Analysis and Result

#### 3.2.1. Hierarchical Analysis

A hierarchical structure model by using the analytic hierarchy process was established, and 25 experts were asked to score on it. After the statistics, those expert opinion results that did not conform to the consistency of the matrix and the records with serious data deficiency were removed. Finally, a total of 18 experts’ of the Pairwise evaluation matrix were used.

Using the sort vector synthesis method to calculate the contribution weight of each risk factor to the food quality safety risk value: Let $w^{\left(k\right)}={\left(w_{1}{}^{\left(k\right)},\;w_{2}{}^{\left(k\right)},\;\ldots ,\;w_{n}{}^{\left(k\right)}\right)}^{T}\;k=1,\;2,\;\ldots ,\;m$ be the sorting vector obtained by judging the eigenvalue of the matrix by the k expert, and then record the average integrated sorting vector as $w={\left(w_{1},\;w_{2},\;\ldots ,\;w_{n}\right)}^{T}.\mathrm{Expert}\;\mathrm{weight}\;\lambda_{1},\;\lambda_{2},\;\ldots ,\;\lambda_{n}=\frac{1}{m}\geq 0,\;\mathrm{meat}\;\overset{m}{\underset{j=1}{\sum }}\lambda_{j}=1,\;\mathrm{calculated}$:(15)$$\;w_{j}^{\prime }={\left(w_{j}^{\left(1\right)}\right)}^{\lambda_{1}}{\left(w_{j}^{\left(2\right)}\right)}^{\lambda_{2}}\ldots {\left(w_{j}^{\left(m\right)}\right)}^{\lambda_{m}}\;j=1,\;2,\;\ldots ,\;n\;$$

Normalized to obtain the average integrated ranking vector, and further calculating the contribution weight of each risk factor to the food quality safety risk value: W = [0.137455964, 0.108826553, 0.040800586, 0.117854333, 0.024376977, 0.023713985, 0.01271776, 0.014183814, 0.010006234, 0.009234366, 0.020952994, 0.017363897, 0.004244048, 0.034688485, 0.139479073, 0.214638755, 0.032430207, 0.037031968].

#### 3.2.2. Distribution Fitting Analysis

The distribution fitting of processed data with @risk can be divided into four steps: Define input data, select the distribution type to be fitted, run the fitting, test the fitting degree, and select the most suitable distribution. Taking the policy adjustment risk as an example, the simulation results show that the overall quality safety risk obeys the Beta General distribution: Risk Beta General (3.5954, 23.116, 6.3774, 126.35).

Both p-p and q-q graphs are approximately straight lines, indicating a good fitting, as shown in Figure 1. Then, the chi-square test was performed. When α = 0.05, the critical value obtained by the table reference was 41.3371, and the chi-square value of quality safety risk distribution was 20.3360, which is less than the critical value. Moreover, the *p*-value was 0.8519, indicating that this probability distribution was acceptable. The simulation results are shown in Figure 2. The average food quality safety risk value of the DLS group is 22.5257. The minimum risk value is 7.2260 and the maximum risk value is 56.0929. At the same time, the probability that the quality safety risk value is less than 10 is 1.3%. At 41.4% probability, the value is more than 10 and less than 20; at 41.4%, the value is more than 20 and less than 30; at 12.9%, the value is more than 30 and less than 40; at 2.6% probability, the value is more than 40 and less than 50; at 0.4% probability, the value is greater than 50 and less than 60.

Since the DLS is an enterprise with very high requirements in itself, the threshold value of all critical control points is lower than even the most stringent international standards, including the European Union regulations and Japanese affirmation list. According to the simulation results, the maximum value of its overall risk value is only 56.0929. After communication with the internal auditor for the enterprise risk, it is tentatively determined that the enterprise does not want the total risk value to be higher than 70. It can be seen that the current status of DLS enterprise’s food quality safety risk failed to trigger early warning. Unlike traditional warnings, which think it is safe only when it reaches the threshold, companies can still analyze what might happen and take precautions in advance.

The sensitivity analysis on a different early warning index shows that concern about counterfeiting has the greatest impact on food quality safety risk, with a correlation coefficient of 0.5. It was followed by the policy adjustment risk with a correlation coefficient of 0.43. The source hazards of raw materials ranks third with a correlation coefficient of 0.35. A further scenario analysis shows: If the ideal risk value is 40, the probability of exceeding 40 will be 3%. Although the probability is small, the probability does not mean that it will not happen. The analysis on the risk factors that have a critical impact on the risk value that is over 40 can derive such conclusions and contribute to such topics as “concerned about counterfeiting”, “source hazards of raw materials”, “natural disasters and climate impacts”, “policy adjustment risk”, “economic environment risk”, “product innovation risk”. If you want to control the overall risk value that is not more than 40, then you need to focus on the six elements of control.

## 4. Conclusions

The risk early warning index established takes into account on all the certification systems and standards required to the meat product processing enterprises. It combines the enterprise internal and external various factors that may affect the food quality; as well as expands the traditional meat products enterprise quality control index system. It proposed the risk early warning index that includes enterprise external environmental risk, internal environmental risk, and risk of consumers concern.

It shows that the food quality safety risk of enterprises in the meat processing industry is a probability distribution through the Monte Carlo simulation. This method not only compares the risk value with the ideal situation, but also compares it with different situations. 

The case study of DLS was done considering the six important risk early warning indexes that were given with reference to the enterprises practice. In order to control the overall risk, it is suggested that enterprises should add six measures into their critical control points system, and set up control points inside the customer service department, incoming inspection, the workshop temperature and humidity, the marketing department, and R&D department, respectively. 

Due to the limitations of time and conditions, the prediction problem has not been included. It is believed that further studies can apply historical data to complete early warning and prediction in order to make early warning more valuable.

## Figures and Tables

**Figure 1 ijerph-17-06579-f001:**
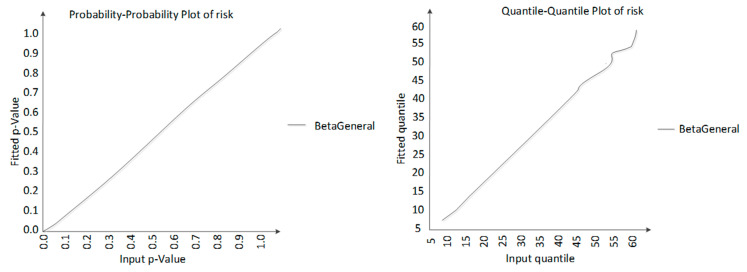
P-P Plot (Probability-Probability Plot) and Q-Q Plot (Quantile-Quantile Plot).

**Figure 2 ijerph-17-06579-f002:**
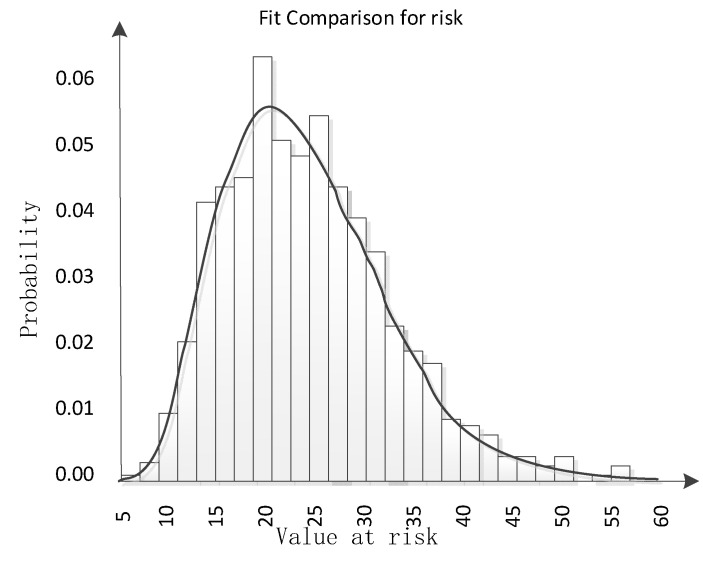
The simulation results.

**Table 1 ijerph-17-06579-t001:** Food quality and safety risk warning indicators of meat processing industry.

The Target Layer	First-Level Index	The Secondary Indicators
Risk of food quality safety (A)	External environmental risk (B1)	Policy adjustment risk (u1)
		Economic environment risk (u2)
		Natural disasters and climate impacts (u3)
		Source hazards of raw materials (u4)
	Internal environmental risk (B2)	Insufficiency of top management responsibility (u5)
		Personnel management risk (u6)
		Hazard control risk (u7)
		Product innovation risk (u8)
	Risk of consumers concern (B3)	Concerned about the harmful substances (u9)
		Concerned about counterfeiting (u10)
		Concerns about sensory quality (u11)
		Concerns about the environment (u12)

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
