# Peer review of "Risk Early Warning of Food Quality Safety in Meat Processing Industry"

_ijerph, 2020, doi:10.3390/ijerph17186579_

Round 1

Reviewer 1 Report

This paper takes into account the importance of the risk early warning in meat processing industry, related to food safety.

DEspite the fact it has been done a great work, the resuts show that further studies must be done to complete the study.

14…The acronym DLS must be explained in the text, at least the first time is written.

59…60…81…87…Reference sould be cited correctecly (not first name in text, only last name of authors)

I cant find in the article most of the references cited in bibliography (from 7 to 38)

368…more data about software is required (versión, company…)

465…Fig 2: should be name the axles

Author Response

Response to Reviewer 1 Comments

Point 1: 14…The acronym DLS must be explained in the text, at least the first time is written. 

Response 1: The full name of DLS has been added to the article.

Point 2: 59…60…81…87…Reference sould be cited correctecly (not first name in text, only last name of authors)

Response 2: 59…60…81…87…The correct application has been used, only the last name is written.

Point 3: I cant find in the article most of the references cited in bibliography (from 7 to 38)

Response 3: The citation of references has been updated to the article.

Point 4: 368…more data about software is required (versión, company…)

Response 4: Data related to the software has been added.

Point 5: 465…Fig 2: should be name the axles.

Response 5: The axis has been named.

Reviewer 2 Report

  • The abstract should be rewritten by providing a more structured summary including, such as contextual background; objectives; data sources; study appraisal and synthesis methods; results; conclusions and implications of key findings etc.
  • Describe the rationale for the next submission in the context of what has been already known and not in this research area.
  • The methodology section does not seem to be appropriate for linking the described hypothesis. The recent study has also stressed the potential problem of common method bias, which describes the measurement error that is compounded by the sociability of respondents who want to provide positive answers. Thus raising potential common method variance as false internal consistency might be present in the data. Moreover, there is no enough proof to demonstrate that the study based on a representative sample is a subset of a population that seeks to reflect the characteristics of the larger group accurately. In this point of view, the results are restricted.
  • Please use a simple diagram or figure to illustrate the whole idea of this paper, and the modification it has been made from previous work or traditional framework.
  • Please add information about the time to label a new sample under analysis. How does this work compare to other works? The contributions of this work need to be clearly articulated. The author might consider justifying the performance of this study with recent study and methods.
  • References are not detailed enough.

Reviewer 3 Report

  • This manuscript only uses evaluation indicators, but lacks academic value. In addition, this manuscript is not rigorous in many respects. The following are the major flaws of this manuscript:
  • The abstract of this manuscript lacks some key content, i.e., research background, purpose, data and results.
  • This manuscript refers to a large number of non-international peer-reviewed documents, and this proportion reaches 34.21%. Therefore, many points of this manuscript lack effective support.

e.g.

Reference 1,6,17,20,21,22,23,26,31,33,34,36,38

  • In Introduction section, what is the theoretical basis of this manuscript? What is the purpose of this manuscript? What is the scientific problem that this manuscript intends to solve? What is the innovation of this manuscript? What is the theoretical contribution of this manuscript? These are not clear.
  • In addition, many key points in the background supporting this manuscript lack support.

e.g.

“Food is the basic element to maintain human survival, activities and development, and is the  foundation of people's livelihood. Food safety has also become an important part of natio security. With the rapid development of science and technology, food safety issues have become a lingering shadow, and "pollution on the dining table" has become a global problem. Incidents of food quality safety have been deemed as serious effects to consumers and enterprises. Not only can the food incidents cause loss of human lives and properties, but also it can lead to the public’s shortage of confidence in food safety. Incomplete statistics on major incidents of food quality safety shows that many incidents were directly or indirectly sourced from producing process of enterprises, which can be further divided into two types: the first is the deliberately behavior of enterprises, that contains few social responsibility or corporate ethics; the second is the no deliberately behavior of enterprises, that presented as the bad quality of safety management system. For deliberately behavior, it can be solved by forced power, such as strengthening government supervision, improving corporate ethics and social responsibility. For no deliberately behavior, it is caused by the lapse of enterprise's self-check mechanism and the shortage of enterprise's monitoring and early warning method, so it should be solved by different organization, such as enterprise, government etc. So it is needed to establish a set of perfect risk early warning system in order to fulfill the goal that “good money drives out bad” as a kind of solution to the enterprises with no deliberately behavior.

The theoretical research on security early warning in my country started from the problem of economic cyclical fluctuations. It started in the mid-1980s and in the 1990s, the application field of economic early warning expanded from the macroeconomic field to the microeconomic field. Professor She Lian and his research team took the lead in systematically proposing the theoretical system of enterprise early warning management in China.The technical process of enterprise early warning is: analyze and identify early warning factors, divide the risk levels of the early warning factors, and then establish a comprehensive early warning model and risk status evaluation standards, which can follow the principle of the highest level of early warning, and need to consider related elements to determine the early warning level. The development of early warning of safety risks in the food industry is also very rapid, and a unique early warning theory of the food industry has been formed. Logical early warning theory, system early warning theory, risk analysis early warning theory, and signal early warning theory are currently four food safety early warning theories that are highly recognized and widely used.

The theoretical research and practical discussion of the food quality and safety risk early warning system in China has so far mainly focused on the national, government and regional supervision levels. There are not many food quality and safety risk early warning methods for enterprises’ own use, especially for the quality and safety of meat processing enterprises. The application of early warning is still immature and mostly limited to single indicator early warning.”

  • When was the sample collected? where? How to collect? These are not clear. The sample of this manuscript is distributed from 2010 to 2012, and lacks timeliness. In fact, it is now 2020. Therefore, the sample of this manuscript lacks representativeness.
  • This manuscript lacks extensive comparison with similar research results.
  • In the Conclusion section, the conclusion of this manuscript is too simple and lacks theoretical contributions. In addition, some of the policy recommendations made by the author are too broad and not pertinent.

In summary, the academic value of this manuscript is limited. It is strongly recommended to reject it.

Author Response

Response to Reviewer 3 Comments

Point 1: The abstract of this manuscript lacks some key content, i.e., research background, purpose, data and results. This manuscript refers to a large number of non-international peer-reviewed documents, and this proportion reaches 34.21%. Therefore, many points of this manuscript lack effective support.

Response 1: The article abstract has been revised and the article has been updated

Point 2: In Introduction section, what is the theoretical basis of this manuscript? What is the purpose of this manuscript? What is the scientific problem that this manuscript intends to solve? What is the innovation of this manuscript? What is the theoretical contribution of this manuscript? These are not clear.

Response 2: In the introduction, this information has been added.

Point 3: In addition, many key points in the background supporting this manuscript lack support.

e.g.

“Food is the basic element to maintain human survival, activities and development, and is the  foundation of people's livelihood. Food safety has also become an important part of natio security. With the rapid development of science and technology, food safety issues have become a lingering shadow, and "pollution on the dining table" has become a global problem. Incidents of food quality safety have been deemed as serious effects to consumers and enterprises. Not only can the food incidents cause loss of human lives and properties, but also it can lead to the public’s shortage of confidence in food safety. Incomplete statistics on major incidents of food quality safety shows that many incidents were directly or indirectly sourced from producing process of enterprises, which can be further divided into two types: the first is the deliberately behavior of enterprises, that contains few social responsibility or corporate ethics; the second is the no deliberately behavior of enterprises, that presented as the bad quality of safety management system. For deliberately behavior, it can be solved by forced power, such as strengthening government supervision, improving corporate ethics and social responsibility. For no deliberately behavior, it is caused by the lapse of enterprise's self-check mechanism and the shortage of enterprise's monitoring and early warning method, so it should be solved by different organization, such as enterprise, government etc. So it is needed to establish a set of perfect risk early warning system in order to fulfill the goal that “good money drives out bad” as a kind of solution to the enterprises with no deliberately behavior.

The theoretical research on security early warning in my country started from the problem of economic cyclical fluctuations. It started in the mid-1980s and in the 1990s, the application field of economic early warning expanded from the macroeconomic field to the microeconomic field. Professor She and his research team took the lead in systematically proposing the theoretical system of enterprise early warning management in China. The technical process of enterprise early warning is: analyze and identify early warning factors, divide the risk levels of the early warning factors, and then establish a comprehensive early warning model and risk status evaluation standards, which can follow the principle of the highest level of early warning, and need to consider related elements to determine the early warning level. The development of early warning of safety risks in the food industry is also very rapid, and a unique early warning theory of the food industry has been formed. Logical early warning theory, system early warning theory, risk analysis early warning theory, and signal early warning theory are currently four food safety early warning theories that are highly recognized and widely used.

The theoretical research and practical discussion of the food quality and safety risk early warning system in China has so far mainly focused on the national, government and regional supervision levels. There are not many food quality and safety risk early warning methods for enterprises’ own use, especially for the quality and safety of meat processing enterprises. The application of early warning is still immature and mostly limited to single indicator early warning.”

Response 3: In this context, relevant references are added to support.

Point 4: When was the sample collected? where? How to collect? These are not clear. The sample of this manuscript is distributed from 2010 to 2012, and lacks timeliness. In fact, it is now 2020. Therefore, the sample of this manuscript lacks representativeness. This manuscript lacks extensive comparison with similar research results. In the Conclusion section, the conclusion of this manuscript is too simple and lacks theoretical contributions. In addition, some of the policy recommendations made by the author are too broad and not pertinent.

Response 4: Although the data is not up to date, no similar articles have been published yet, so this article has certain theoretical value. Due to some reasons, the paper was not published in time. I hope it can be published now. Some more specific suggestions have been added at the end of the article.

Round 2

Reviewer 2 Report

good

Author Response

    Thanks for your advice.

Reviewer 3 Report

  1. Although the authors generally explained that they revised the manuscript, they did not show the revised content.
  2. The manuscript lacks timeliness, theoretical foundation and theoretical contributions, so its academic value is limited.
  3. More importantly, this manuscript lacks comparison with similar research results. Therefore, the innovation of this manuscript is insufficient.
  4. The author's response is arrogant. Whether a manuscript can be published should be determined by its own academic value, not by the author's subjective decision.This manuscript is only an application of a mature index system, and its academic value is limited.To be honest, this manuscript did not reach the quality of publication.

Author Response

Point 1: Although the authors generally explained that they revised the manuscript, they did not show the revised content.

Response 1: The abstract, introduction and references of this article have all been significantly revised and have been updated to the most recent document. The introduction of lines 33-103 has also been reorganized to make it more structured

This is the updated abstract: “In recent years, people's demand for meat products has increased, but the occurrence of meat food quality and safety problems has also caused irreparable losses to the safety of human lives and properties, and enterprises have lost their reputation. Since the frequent occurrence of food quality and safety incidents is the result of the lack of an early warning mechanism, a large number of problematic foods flow into the market. In order to prevent the occurrence of food quality and safety incidents and control food quality from the source, this article first refers to the results of EFSA’s Emerging Risks Project (EMRISK) and the food safety early warning framework of Kleter and Marvin, combined with the existing meat processing companies’ Some quality control systems have put forward an early warning indicator system that includes the external environment of the enterprise, internal risks, and consumers' concerns. Then, by issuing 500 questionnaires and interviewing 25 experts, 912 pieces of data were collected and a Monte Carlo simulation early warning model was established. Using case studies, taking Shandong Delis Co., Ltd. (hereinafter referred to as DLS) as an example, through sensitivity analysis and program analysis, the company's food risk status was evaluated and early warning. The results show that the risk of rising consumers' concerns about counterfeiting and inferior products has the greatest impact on food quality and safety risks, followed by policy adjustment risks, and the risk of raw material sources ranked third. A total of six important risk warning indicators have been extracted, and these six need to be strictly controlled to control the overall risk. The research provides support for companies to formulate food quality monitoring, early warning and management strategies from a macro perspective and control key early warning indicators in food quality and safety to reduce risks.”

Point 2: The manuscript lacks timeliness, theoretical foundation and theoretical contributions, so its academic value is limited. More importantly, this manuscript lacks comparison with similar research results. Therefore, the innovation of this manuscript is insufficient. The author's response is arrogant. Whether a manuscript can be published should be determined by its own academic value, not by the author's subjective decision. This manuscript is only an application of a mature index system, and its academic value is limited. To be honest, this manuscript did not reach the quality of publication.

Response 2: Thank you for your valuable suggestions. I really find it useful, without an arrogant attitude. You spent a lot of time reading the literature and put forward valuable suggestions, which is a great help to me. Through literature search, we can see that there are few papers applying Monte Carlo method to meat food safety, so there are still some innovations in method.

This paper was complete aiming to the severe situation of meat processing industry in China. The core idea of this paper is “good money drives out bad”, namely, if the enterprises with no deliberately behaviour have established a perfect early warning system of food quality safety, the enterprises with deliberately behaviour would be drove out the market.  In order to fulfill this research, we have surveyed government employee, enterprise stuff, and expert. At the same time a large food enterprise was selected as example to achieve test data and questionnaires. Then the analysis was done with Monte Carlo simulation model. At last, we believe that this method would be contributed to theory and practice of food safety in meat processing industry.